# The role of species ecology in predicting *Toxoplasma gondii* prevalence in wild and domesticated mammals globally

Amy G. Wilson[1]*, David R. Lapen[2], Jennifer F. Provencher[3], Scott Wilson[1,4]

**1** Department of Forest and Conservation Sciences, University of British Columbia, Vancouver, British Columbia, Canada, **2** Agriculture and Agri-Food Canada/Agriculture et Agroalimentaire Canada, Ottawa Research Development Centre, Ottawa, Ontario, Canada, **3** Ecotoxicology and Wildlife Health Division, Environment and Climate Change Canada, National Wildlife Research Centre, Ottawa, Ontario, Canada, **4** Wildlife Research Division, Environment and Climate Change, Pacific Wildlife Research Centre, Delta, British Columbia, Canada

* amy.wilson@ubc.ca

**Data Availability Statement:** All data is available in Dryad: doi:10.5061/dryad.c59zw3rfp.

## Abstract

Macroecological approaches can provide valuable insight into the epidemiology of globally distributed, multi-host pathogens. *Toxoplasma gondii* is a protozoan that infects any warm-blooded animal, including humans, in almost every habitat worldwide. *Toxoplasma gondii* infects its hosts through oocysts in the environment, carnivory of tissue cysts within intermediate host prey and vertical transmission. These routes of infection enable specific predictions regarding the ecological and life history traits that should predispose specific taxa to higher exposure and, thus infection rates of *T. gondii*. Using *T. gondii* prevalence data compiled from 485 studies representing 533 free-ranging wild mammalian species, we examined how ecological (habitat type, trophic level) and life history (longevity, vagility, gestation duration and torpor) traits influence *T. gondii* infection globally. We also compared *T. gondii* prevalence between wild and domesticated species from the same taxonomic families using data compiled from 540 studies of domestic cattle, sheep, and pigs. Across free-ranging wildlife, we found the average *T. gondii* prevalence was 22%, which is comparable to the global human estimate. Among ecological guilds, terrestrial species had lower *T. gondii* prevalence than aquatic species, with freshwater aquatic taxa having an increased prevalence compared to marine aquatic species. Dietary niches were also influential, with carnivores having an increased risk compared to other trophic feeding groups that have reduced tissue cyst exposure in their diet. With respect to influential life history traits, we found that more vagile wildlife species had higher *T. gondii* infection rates, perhaps because of the higher cumulative risk of infection during movement through areas with varying *T. gondii* environmental loads. Domestic farmed species had a higher *T. gondii* prevalence compared to free-ranging confamilial wildlife species. Through a macroecological approach, we determined the relative significance of transmission routes of a generalist pathogen, demonstrating an increased infection risk for aquatic and carnivorous species and highlighting the importance of preventing pathogen pollution into aquatic environments. *Toxoplasma gondii* is increasingly understood to be primarily an anthropogenically-associated pathogen whose dissemination is enhanced by ecosystem degradation and human subsidisation of free-

**Funding:** AW was financially supported by Agriculture and Agri-Food Canada's Environmental Change One Health Observatory (ECO2) [J-002305.001.07] to DRL. The funders had no role in study design, data collection and analysis, decision to publish, or preparation of the manuscript.

**Competing interests:** The authors have declared that no competing interests exist.

roaming domestic cats. Adopting an ecosystem restoration approach to reduce one of the world's most common parasites would synergistically contribute to other initiatives in conservation, feline and wildlife welfare, climate change, food security and public health.

## Author summary

*Toxoplasma gondii*, a relative of malaria, is one of the most common parasitic infections in the world, capable of causing severe and chronic health problems to host species, which encompasses all warm-blooded animals. Humans, wildlife and domestic animals all suffer from this pathogen, which is contracted through contaminated water, food and *in utero*. To reduce the global burden of this parasite, we must understand the most common routes of infection and which species will be most likely to be infected. To answer this, we compiled infection data from 950 studies of free-ranging and domestic farm animals. We found higher infection rates in larger predator species, wildlife living in freshwater, and species travelling over larger distances. Higher infection rates in these types of wildlife species highlight the importance of water contamination and the higher risk to predator species, many of which are undergoing serious population declines. We also found that domestic farm animals have higher infection rates than closely related wildlife species, likely caused by their exposure to farm cats. Domestic free-roaming cats are the most significant contributors to environmental contamination and infection of food animals due to their large populations and frequent encroachment into natural and farm areas. Effective, multi-faceted approaches must be adopted to manage free-roaming cat populations to protect wildlife and human health, food security and feline welfare.

## Introduction

Pathogens can profoundly impact wildlife populations through overt or more insidious effects on health [1,2]. The impacts of pathogens on wildlife can be exacerbated via the interaction with other stressors such as habitat destruction, food scarcity, climate change, invasive species and toxicant exposure, making it a conservation priority to predict the species or populations that are disproportionately threatened by pathogen-mediated declines [1,2]. Generalist pathogens with domestic animal reservoirs that affect multiple species, including humans, are particularly poignant research targets, given the relevance for human, wildlife and domestic animal health [3,4]. Insight into the most consequential transmission pathways of generalist pathogens with multiple routes can be gained using a macroecology approach of comparing prevalence patterns among species that vary in ecological traits. *Toxoplasma gondii* is a zoonotic protozoal pathogen with significant conservation and global health significance that lends itself well to a macroecological approach due to its cosmopolitan distribution and capacity as a generalist pathogen [4].

*Toxoplasma gondii* can infect any endothermic host species, with a third of the human population infected [5]. Hosts are infected environmentally through an oocyst stage, trophically through ingesting a tissue cyst stage (bradyzoite) and *in utero* through a blood-borne stage (tachyzoite). The complete *T. gondii* life cycle involves a felid as the definitive host and any warm-blooded animal acting as an intermediate host. Infected felids excrete oocysts in their feces, infecting any intermediate host that ingests these oocysts via contaminated water or food. Oocyst loads in the environment can reach high levels because felids excrete millions of oocysts over multiple episodes throughout a felid's lifespan in response to *T. gondii* exposure

or re-exposure [6]. These oocysts are resilient to environmental conditions with long-term survival [7,8], dispersing passively through soil/water transport pathways [9]. If the intermediate host survives the initial infection, *T. gondii* encysts within the tissues of the infected individual, persisting as a dormant infection. If any felid or non-felid predator consumes the infected tissues of an intermediate host, the predator is then infected [5]. The ability for *T. gondii* to be infective between intermediate hosts through carnivory, and not just between definitive and intermediate hosts, is unique to *T. gondii* and not present in other related pathogens [10]. Trophic transmission between intermediate hosts allows for long-term maintenance of *T. gondii* in the environment and enables *T. gondii* infections to occur without felids [10].

The consequences of a *T. gondii* infection or reactivation depend on the host individual's immune status and *T. gondii* strain [5,11]. In immunocompromised hosts, *T. gondii* can cause fatalities through meningoencephalitis, pneumonitis, myocarditis and hepatitis [11]. Acute or reactivated *T. gondii* infections during pregnancy can lead to miscarriages, fetal death, and severe congenital defects such as hydrocephalus, blindness and intellectual disability [11]. Although immunocompetent humans with latent *T. gondii* infections may lack overt symptoms, large clinical syntheses have suggested latent toxoplasmosis increases the risk and morbidity of a range of diseases including neurodegenerative, psychiatric, ocular, autoimmune, hepatic, cardiovascular disease and several forms of cancer [12]. Comparable clinical syndromes are reported in food animals, with the most economically relevant effect being reproductive failure [13].

Wildlife necropsy reports and field studies indicate that *T. gondii* infections impact wildlife in similar ways as humans and domesticated animals [5]. Toxoplasmosis has caused high rates of mortality in multiple critically endangered taxa, including birds [14,15], marine mammals [16,17], felids [18], deer [19] and rodents [20]. Wildlife with latent *T. gondii* infections are more likely to suffer mortality from other parasitic co-infections [21], vehicular collisions [22], cold weather [23] and predators [24]. Infected wildlife are also more likely to experience delays in reproductive development [25]. Impacts on reproduction lead to large economic losses in food animal production [26]. Although cattle appear more clinically resistant, sheep and goats are more sensitive, with global estimates of 3–54% of abortions attributable to toxoplasmosis [5,13].

The ability of *T. gondii* to infect hosts through multiple routes and life stages means that species whose ecological traits differentially expose them to certain stages should experience different infection rates. Thus, by examining infection rates relative to ecological and life history characteristics, we can gain important insight into how environmental and life history drivers influence prevalence patterns [27]. Our first objective was to assess the influence of diet and habitat on the prevalence of *T. gondii* in free-ranging mammalian wildlife. Waterborne transmission is a significant exposure route for all mammals, irrespective of diet, due to the risk of oocyst ingestion by drinking water and active foraging [28]. However, aquatic species are immersed in water, placing them at an increased risk of encountering oocysts, which can be distributed throughout the water column, providing a multi-dimensional exposure surface. This risk is exacerbated by the ability of *T. gondii* to concentrate on aquatic particulates, sediment, biofilms and filter-feeding aquatic prey items [9]. Therefore, we predicted that aquatic mammals would have higher prevalence patterns than terrestrial mammals. Since *T. gondii* oocysts enter the aquatic system from terrestrial pollution pathways (i.e., surface runoff and preferential flow) [29], we also predicted that freshwater aquatic species would have an increased risk relative to marine aquatic species. In addition to habitat-associated exposure risks, animals contract *T. gondii* through their diet by ingesting oocysts or tissue cysts. For herbivores, diet-associated infection occurs by ingesting oocyst-contaminated vegetation and soil. Carnivores and omnivores have the additional risk of being infected by consuming infective tissue cysts within intermediate host prey. Carnivores that scavenge may have an even higher

risk of *T. gondii* infection because scavengers that consume higher trophic levels have increased potential for bioaccumulation. Scavenging taxa also often have a predilection for using anthropogenic subsidies, a known risk factor for *T. gondii* [30,31]. Therefore, we predicted that globally, *T. gondii* prevalence would increase with carnivory and scavenging.

Our next objective focused on life history traits that could influence *T. gondii* prevalence, including longevity, gestation duration, hibernation and dispersal distance. *Toxoplasma gondii* prevalence increases with individual age [32,33], and we predicted this age-associated trend could manifest as longer-lived species having a higher prevalence. Similarly, since *T. gondii* can be transmitted *in utero*, we assessed if *T. gondii* prevalence was higher in species with longer gestation duration, given the larger infection window for maternal-fetal transmission. Since *T. gondii* can only successfully infect endothermic species, we tested if mammalian species undergoing torpor or hibernation with the accompanying decreased body temperature [34] and reduced time frame for exposure opportunities had a lower prevalence than ecologically comparable species. Finally, we evaluated how vagility could impact the prevalence of *T. gondii*. There are strong geographic determinants of *T. gondii* infection risk, such as human density [30,31], landscape conversion [35] and free-roaming cat density [29,31]. Therefore, we predicted that highly vagile species would have an increased probability of infection due to the increased probability of travelling through these high-risk locations. Our final objective compared prevalence patterns between domestic animals (cattle, sheep, pigs) and free-ranging wildlife from the same taxonomic family. We predicted domestic animals would have higher infection rates due to previously demonstrated associations between anthropogenic disturbance and *T. gondii* [29–31].

## Results

Our literature search identified 485 studies across 93 countries for free-ranging wildlife species. Across these studies, there were 148, 425 individuals tested for *T. gondii* infection, representing 18 taxonomic orders, 86 taxonomic families and 533 taxonomic species (Fig 1). For domestic species, focusing on cattle (*Bos taurus*), sheep (*Ovis aries*) and swine (*Sus domesticus*), data was compiled from 540 studies across 77 countries and 384, 349 individuals (Fig 1). In general, countries with high testing rates for *T. gondii* in domestic animals also had high testing in wild mammals (Fig 1).

Habitat, diet, and dispersal distance were all significant variables when evaluated individually and were included within a single global model. Terrestrial species had a lower risk of *T. gondii* exposure (β = -0.77; 95% CI: -1.41 --0.18) compared to aquatic taxa. Among aquatic species, marine species had a significantly lower risk of *T. gondii* exposure (β = -1.02; 95% CI: -1.78 --0.23) relative to freshwater aquatic species. Species that are not exposed to tissue cysts through carnivory, such as herbivores, insectivores, invertivores and piscivores, had a significantly reduced risk of *T. gondii* (β = - 1.23; 95% CI: -1.89 --0.55) relative to carnivorous species. Omnivores did not differ significantly from carnivores (β = -0.11; 95% CI: - 0.71–0.46). Among the evaluated life history traits of generation time, gestation length and dispersal distance, only dispersal distance significantly affected *T. gondii* infection risk. Species with larger average dispersal distances had an increased risk of *T. gondii* exposure (β = 0.22; 95% CI: 0.05–0.37). Studies relying on molecular isolation or bioassays had a significantly lower prevalence (β = -0.84; 95% CI: -1.28 –- 0.37) than studies using serology.

There was a significant phylogenetic signal present in the dataset, where phylogenetic variation (λ) accounted for 41% of the random variation (95% CI: 0.21–0.63), while the random variation attributed to species accounted for 8% of total variation (95% CI: 0.03–0.18). For the second analysis, the model including the wild/domestic predictor had greater support (AIC:

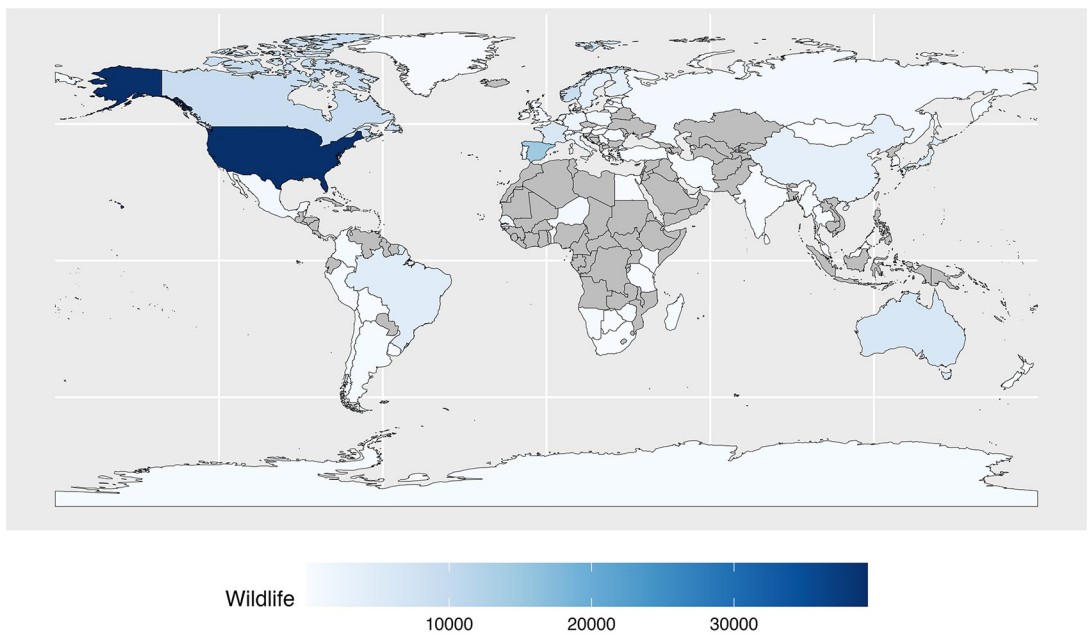

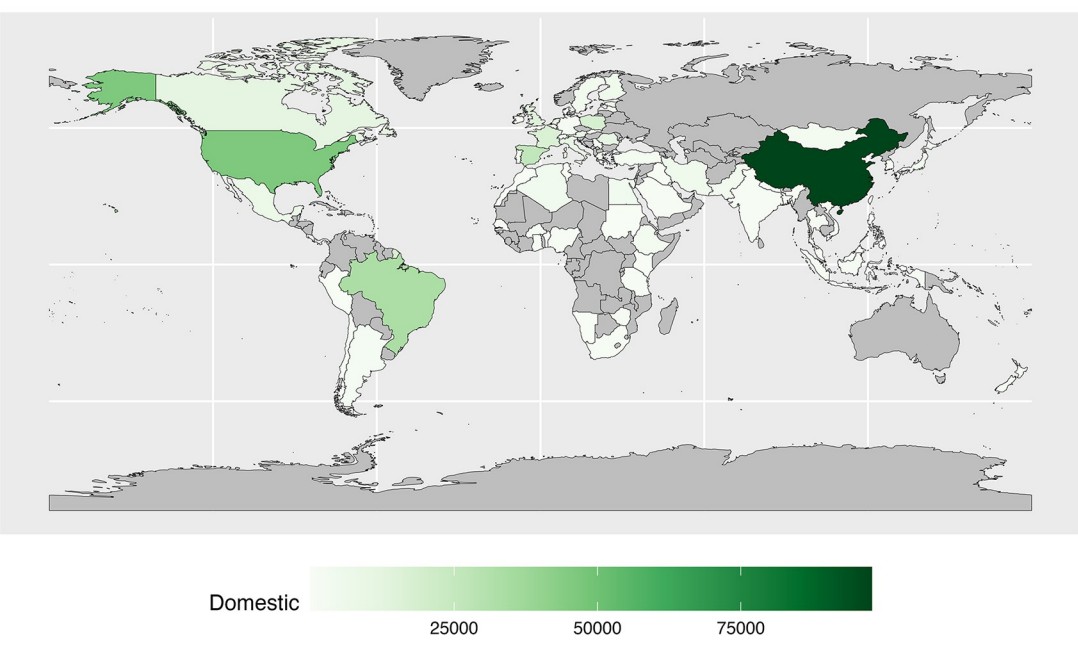

**Fig 1. Distribution of study sites included in the global analysis of *Toxoplasma gondii* prevalence data for domestic animals and free-ranging wildlife.** The country-level sampling intensity in terms of number of individuals sampled is shown for wildlife (blue) and domestic animals (green). Countries for which *T. gondii* prevalence data was not found within the scope of our search are shown in grey. Basemap was made with Natural Earth (https://www.naturalearthdata.com).

9103.1) compared to models with only taxonomic family (AIC: 9110.0) or continent (AIC: 9934.9). In this top model, wild species had a lower risk of *T. gondii* infection relative to the domestic confamilial species. (β = -0.39; 95% CI: -0.02 –- 0.76).

The global prevalence for free-ranging wildlife compiled across all studies was 22%, with substantial variation across taxa. Considering taxonomic families with at least 200 individuals

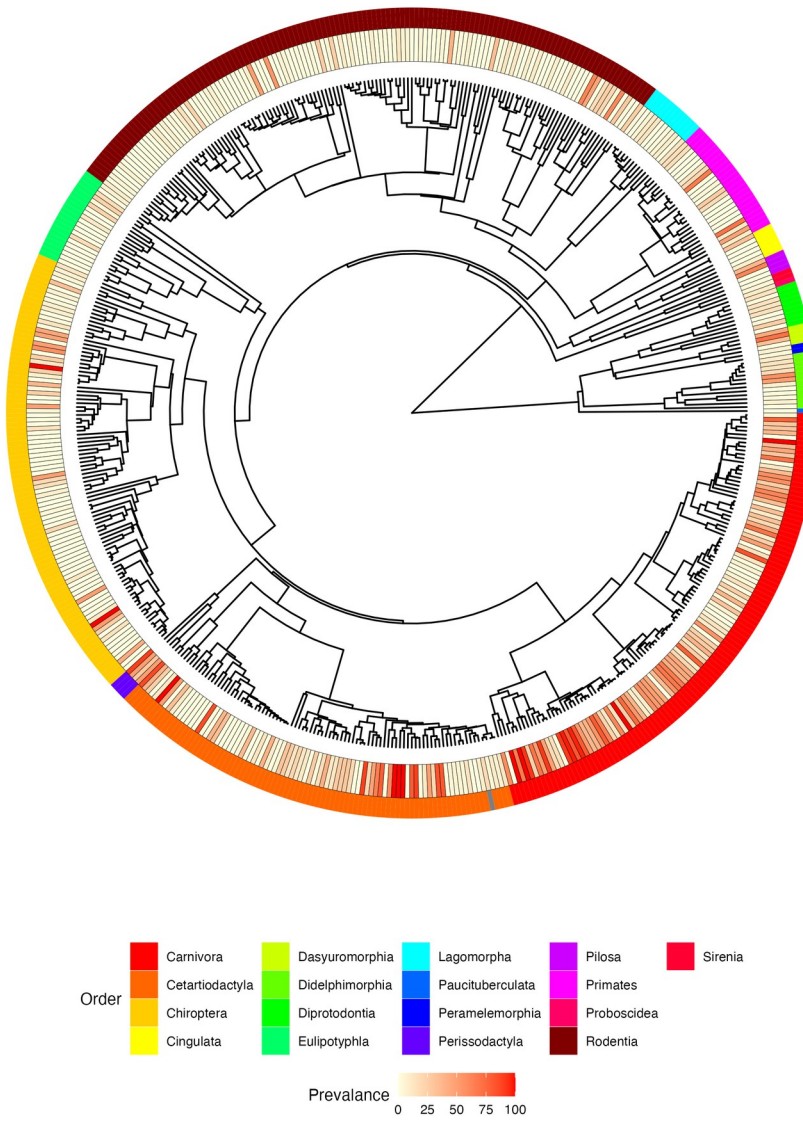

**Fig 2. Node-dated phylogeny of 530 species included in T. gondii analysis.** The maximum clade credibility topology of 10,000 trees is shown. The inner panel depicts the species-level *T. gondii* prevalence over all available studies. The outer panel identifies the taxonomic families.

sampled, some of the lowest prevalences were found in marine mammals and rodents, while prevalences exceed 40% in Felidae, Iniidae, Tayassuidae, Mustelidae, Ursidae and Hyaenidae (Fig 2 and S1 Table). The species-level prevalences were highly variable, and some taxa lacked sufficient sample sizes to obtain meaningful prevalence estimates (S1 Table). Across all studies for domestic animals, the overall *T. gondii* prevalence was 23%. Although wild populations of Bovidae and Suidae had lower average prevalences than domestic populations, there was considerable inter-continental variation (S2 Table).

## Discussion

Using a large global database, we demonstrated that *T. gondii* prevalence is influenced by species ecological traits consistent with *T. gondii* epidemiology and ecology. The contribution of

tissue cyst-associated infections towards maintaining *T. gondii* within the environment was apparent by the significant influence of carnivory on prevalence in wildlife populations. Carnivores and omnivores ingesting tissue cysts can be exposed to thousands of highly infectious *T. gondii* bradyzoites, which, at least in cats, were more infectious than oocysts [5]. This additional transmission route with a larger dose of a potentially more infectious life stage explains the higher prevalence for carnivorous and omnivorous species that we found in this global study and has also been shown in other sympatric comparisons [36,37] and syntheses [27]. Terrestrial omnivorous species are often more synanthropic [38], which would increase exposure risk, even if the rate of carnivory is lower than true carnivores. We also found no effect of scavenging on *T. gondii* prevalence, suggesting the bradyzoites within tissue cysts may rapidly lose infectivity. Aquatic carnivores are primarily piscivores and invertivores, meaning that diet-associated infections would be disproportionately oocyst-associated because fish, invertebrates and mollusks can accumulate oocysts but do not form tissue cysts [39,40].

Despite the reduced exposure to tissue cysts and the absence of an aquatic definitive host, we found that aquatic species had a higher *T. gondii* prevalence compared to terrestrial species. Exposure in the aquatic environment can occur throughout the water column, increasing the risk that an individual could inadvertently ingest an oocyst. Oocysts also concentrate on aquatic particulates [41], microorganisms [42], biofilms [41], and even microplastics [43], increasing the infective dose and elevating infection risk for aquatic taxa. Freshwater taxa are considered particularly at risk due to the more direct and frequent exposure to contaminated terrestrially-based waters without any dilutive effect associated with open ocean. In this study, species associated with freshwater environments had an increased *T. gondii* prevalence. Increased *T. gondii* prevalence associated with polluted water exposure has been shown in muskrats (*Ondatra zibethicus*) [44] and Yellow-legged gulls (*Larus michahellis*) [45]. Variation in exposure to terrestrial runoff has also been positively associated with *T. gondii* prevalence in marine species such as sea otters (*Enhydra lutris*) [46] and beluga (*Delphinapterus leucas*) [47]. The increased *T. gondii* prevalence in aquatic mammalian species in this study supports earlier suggestions of the importance of waterborne toxoplasmosis in the epidemiology of *T. gondii* [28].

Certain life history traits could also differentially predispose particular species to infection rates from generalist pathogens. We found an important positive relationship between dispersal distance and *T. gondii* prevalence but no relationship between *T. gondii* and longevity, gestation duration or hibernation behaviour. Species with higher vagility may have an increased prevalence due to the increased probability of travelling through an area with a high *T. gondii* infection risk. Factors that are associated with increased *T. gondii* prevalence in wildlife are exposure to urbanization [31,48–50], agriculture [51], sewage [52,53], anthropogenic food provisioning [54] and domestic cat abundance [29–31]. Synanthropic wildlife often suffer from higher pathogen loads [55], but as we were unaware of any indices of synanthropism for wildlife taxa, we could not test that explicitly.

*Toxoplasma gondii* cannot invade and proliferate host tissues when body temperatures are lower than 37°C [56]. However, we were unable to detect any protective influence of hibernation or torpor against *T. gondii* infection, which could be attributable to the absence of an effect, that any effect is small compared to other influential ecological traits, or our sample size was too small to tease out the effects of these two traits. Similarly, contrary to our expectation, we did not find a higher *T. gondii* prevalence in species with a longer gestation duration, suggesting that vertical transmission may have a reduced role in maintaining *T. gondii* within wildlife populations. Although vertical transmission may not be consequential for *T. gondii* persistence, *in utero* transmission could impact wildlife populations through high fetal mortality, which would go unmeasured in most populations and obscure any relationship between

gestation traits and prevalence. Additionally, rodents infected congenitally with *T. gondii* may be unable to mount an immunologic response and would be seronegative [57], further obscuring the importance of vertical transmission. These false negatives could explain the unexpectedly low *T. gondii* prevalence within Muridae, which is thought to be a major intermediate host prey for felids. We also expected longevity to be associated with increased prevalence because multiple intraspecific studies have found *T. gondii* prevalence to increase in older age cohorts [32,33]. However, generation length as a proxy for average longevity was not significantly associated with *T. gondii* prevalence in this study. The absence of a relationship could be influenced by the suitability of generation time as a proxy for longevity, age-specific mortality and morbidity, seroreversion and the inability to distinguish multiple infection events from single infections.

We did find that studies using serosurveys were generally associated with a higher prevalence. Although serological tests, especially the modified agglutination test, have been widely used in wildlife, these tests are rarely validated, such that false positives and negatives are possible [5]. Similarly, molecular or isolation methods require that bradyzoites or tachyzoites are included in the sampled tissue from infected individuals. In cases where infected individuals have low numbers of tissue cysts or in larger-bodied animals, there is an increased risk of sampling error, leading to an underestimation of prevalence. However, if the sample includes the life stage, molecular methods can have high sensitivity and specificity [5]. Although our model controlled for methodological differences, the presence of a phylogenetic signal suggests that our models did not capture some phylogenetically conserved but influential ecological or physiological traits, which may also include taxon-specific differences in method sensitivity and specificity.

Further evidence of anthropogenic determinants of *T. gondii* infection comes from our result that domestic animals had an increased prevalence compared to free-ranging wildlife confamilial species. The presence of free-roaming cats on farms is a demonstrated risk factor for *T. gondii* in food animals [58–60,13], with even low cat densities leading to substantial soil contamination [51,58,59,61]. Outreach programs should ensure that food animal producers are aware that not only are free-roaming cats ineffective for rodent control [62], but this practice increases farm animal infection rates [58,63] by amplifying the *T. gondii* infection cycle through repeated sheds and host prey infections [6]. Integrative pest management strategies should instead recruit local ecosystem services shown to be effective for rodent control, such as attracting native carnivores [62,64] or habitat manipulation [65]. These ecosystem-based interventions may also synergistically benefit other aspects of farm productivity and sustainability.

In this study, we demonstrated that a macroecological approach can help elucidate consequential transmission routes of a generalist pathogen for mammalian wildlife, enabling the development of actionable recommendations. The elevated risk in freshwater aquatic species suggests that prioritising the reduction of pathogen pollution from free-roaming domestic cats for wildlife and food animals will be key for mitigating *T. gondii* infections. Domestic cats are the most consequential definitive host for *T. gondii* [29], elevating the prevalence of *T. gondii* beyond levels imparted by wild felids [29,31]. Strategies that prevent parasite spillover from domestic definitive hosts to wildlife can be highly effective, as demonstrated in the *Trichinella spiralis*-domestic pig cycle [66]. Although some studies call for *T. gondii* mitigation through managing intermediate hosts, given the large environmental oocyst-associated route, this approach would be ineffective given the scale of oocyst contamination and impractical since all warm-blooded animals are intermediate hosts. There are also considerable ethical, economic and legal implications of managing or neglecting native wildlife species instead of managing an introduced species such as domestic cats [67,68]. Focusing on reducing free-roaming cat populations synergistically addresses challenges such as other cat-associated diseases of zoonotic, economic and conservation significance [26,69,70], the substantial wildlife mortality

due to cat-associated predation [71] and the current marginalisation of free-roaming owned and unowned cats [72].

*Toxoplasma gondii's* close association with anthropogenic activities [73] makes it an ideal model pathogen for demonstrating the benefit of a multifaceted ecosystem-level approach towards mitigating important and prolific pathogens globally [4]. For example, restoring landscape features that sequester *T. gondii* oocysts can prevent the transfer and contamination to other downstream receptors of environmental and public health significance [35]. Habitat restoration also provides predation refugia, bolsters wildlife health, and improves reproductive success, all leading to increased disease resiliency within wildlife populations [74]. Ensuring robust populations of apex predators can provide critical ecosystem services by limiting the encroachment of introduced mesopredators, including domestic cats [22,75], reducing predation pressure and opportunities for zoonotic disease spillover. Holistic ecosystem protection is a cost-effective, straightforward, but currently underused approach for disease prevention despite the large economic impact of zoonotic disease and synergistic benefits to biodiversity conservation, climate change mitigation efforts and human health [76].

## Materials and methods

### Literature review

Our analyses first focused on *T. gondii* prevalence studies within populations of free-ranging wild mammals (Class: Mammalia). We searched for studies using Web of Science, PubMed and Google Scholar using search terms: "wild*", "mammal" and "toxoplasmosis" or "toxoplasma." We were able to locate additional studies from the reference lists of these studies, studies citing these works, and comprehensive review articles [5]. For the wildlife analyses, we excluded studies that did not report results from all samples tested, data with only genus-level identifiers, and studies from captive or farmed wild animals. We extracted all data from each study, including the number of infected animals, the total number of animals tested, and the methodology used. For each species, we then compiled the ecological information as described below. We used a similar search protocol to compile prevalence data on cattle (*Bos tarus*), sheep (*Ovis aries*) and swine (*Sus domesticus*) (S3 Table).

### Ecological and life history factors

To test our hypotheses on the ecological drivers of *T. gondii* prevalence in wildlife, we collected ecological and life history information for all taxa included in the compiled dataset. We standardized the taxonomy and extracted information from the COMBINE database of mammalian traits [77]. Data was used as provided in this database with the exception that we reclassified the dietary groups to reflect the risk of tissue cyst carnivory leading to three categories: 1) Species with a primarily non-tissue cyst diet (e.g. herbivores, insectivores, invertivores and piscivores), 2) Carnivores with primarily endothermic prey and 3) Omnivores which consume a mixed tissue cyst and non-tissue cyst diet. Taxa were classified as terrestrial or aquatic, with aquatic being subclassified into freshwater or marine. Scavengers were any carnivore or omnivore species that had >10% of their diet originating from scavenging. For the life history traits, we included gestation duration, whether they engaged in hibernation or torpor, average dispersal distance and generation time as a proxy for longevity.

### Statistical analysis

Analyses used Bayesian phylogenetic generalized linear mixed models as implemented in the R package MCMCglmm [78], following the methods of Barrow et al. (2019) [79]. Phylogenetic

generalized linear models accommodate phylogenetic covariance matrices, controlling for trait sharing or disease susceptibility due to common ancestry. We included phylogenetic variance as a random effect with the phylogenetic covariance matrix for mammalian taxa based on 10,000 birth–death node-dated trees [80], with the R package *ape* [81]. Prevalence data were modelled as a multinomial distribution of counts of positive and negative individuals. We used a base model for model selection that included a fixed effect of the *T. gondii* detection method (serological or isolation) and four random variables (phylogenetic variance, species, study and country). We used this base model to test the influence of habitat (aquatic or terrestrial) tissue cyst carnivory, longevity, gestation duration and dispersal. To evaluate the influence of freshwater and marine exposure and scavenging, we tested a subset of models that only considered aquatic species and terrestrial species, respectively. Predictors found to be statistically significant individually were retained in a top global model. This top model, containing the significant variables, was used to estimate the mean effects of the fixed predictor variables, the phylogenetic signal ($\lambda$) and non-phylogenetic species' effects. Fixed-effect predictors were considered significant at 95% if the 95% credible interval (CI) did not overlap with zero.

A separate analysis was run for domestic species and the confamilial free-ranging wildlife, where we compared the improvement of model support by adding three fixed predictor variables: taxonomic family, domesticated or wild and continent. All tested models included random effects of locality and study. We also calculated the *T. gondii* prevalence with 95% confidence intervals at the species and family levels with *epiR* [82] for free-ranging wildlife and domestic species. All data is available in Dryad [83].

## Dryad DOI

10.5061/dryad.c59zw3rfp

## Supporting information

**S1 Table. Prevalence of *T. gondii* with 95% confidence intervals estimated at the taxonomic family level for free-ranging wild mammal populations compiled across 485 publications.** Numbers of sampled individuals and species are provided in parentheses.
(DOCX)

**S2 Table. Prevalence of *T. gondii* with 95% confidence intervals estimated from a compilation of 540 published global studies for domesticated and free-ranging populations of Bovidae and Suidae.** The numbers of sampled individuals are provided in parentheses.
(DOCX)

**S3 Table. Reference list of publications for which *T. gondii* prevalence data was available for free-ranging wild and domesticated mammal populations.**
(DOCX)

**S1 Data. Complete dataset for free-ranging wild and domesticated mammal populations for *T. gondii* prevalence data.**
(XLSX)

## Acknowledgments

We gratefully acknowledge the efforts of all authors whose published work allowed us to perform this global analysis. We also thank C Ritland and A Miscampbell for discussing *Toxoplasma* detection methods.

## Author Contributions

**Conceptualization:** Amy G. Wilson, David R. Lapen, Jennifer F. Provencher, Scott Wilson.

**Data curation:** Amy G. Wilson, Scott Wilson.

**Formal analysis:** Amy G. Wilson, David R. Lapen, Scott Wilson.

**Funding acquisition:** David R. Lapen.

**Investigation:** Amy G. Wilson, David R. Lapen.

**Methodology:** Amy G. Wilson.

**Project administration:** Amy G. Wilson, David R. Lapen, Scott Wilson.

**Software:** Amy G. Wilson.

**Supervision:** David R. Lapen, Jennifer F. Provencher, Scott Wilson.

**Validation:** Amy G. Wilson, Scott Wilson.

**Writing – original draft:** Amy G. Wilson, David R. Lapen, Scott Wilson.

**Writing – review & editing:** Amy G. Wilson, David R. Lapen, Jennifer F. Provencher, Scott Wilson.

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
