## [Decision Letter · Decision Letter 0]

9 Nov 2023

Dear Dr Wilson,

Thank you very much for submitting your manuscript "The role of species ecology in predicting Toxoplasma gondii prevalence in wild and domesticated mammals globally" for consideration at PLOS Pathogens. As with all papers reviewed by the journal, your manuscript was reviewed by members of the editorial board and by several independent reviewers. The reviewers appreciated the attention to an important topic. Based on the reviews, we are likely to accept this manuscript for publication, providing that you modify the manuscript according to the review recommendations.

Please respond to the reviewer's comments -- in particular, it is vital that you make all raw and analyzed data and their sources readily available.

Sincerely,

Michael L Reese, PhD

Academic Editor

PLOS Pathogens

James Collins III

Section Editor

PLOS Pathogens

Kasturi Haldar

Editor-in-Chief

PLOS Pathogens

orcid.org/0000-0001-5065-158X

Michael Malim

Editor-in-Chief

PLOS Pathogens

orcid.org/0000-0002-7699-2064

Please respond to the reviewer's comments -- in particular, it is vital that you make all raw and analyzed data and their sources readily available.

Reviewer Comments (if any, and for reference):

Reviewer's Responses to Questions

**Part I - Summary**

Reviewer #1: The authors applied a macroecological approach to elucidate the relative importance of different routes of transmission for Toxoplasma gondii, a globally-distributed parasite capable of infecting nearly all warm-blooded animals including humans, domestic animals and wildlife. Utilizing seroprevalence data from more than 1000 published studies, the ecology of Toxoplasma is described, highlighting the significance of anthropogenic alterations to ecosystems and aquatic systems for increased risk for transmission and exposure.

**Part II – Major Issues: Key Experiments Required for Acceptance**

Reviewer #1: (No Response)

**Part III – Minor Issues: Editorial and Data Presentation Modifications**

Reviewer #1: Below are comments, questions, and suggested edits.

Lines 28 and 99: Toxoplasma oocysts are not “free-living” but can survive for long periods of time in the environment.

Lines 31, 41 and 50: I don’t think that “Susceptibility” is the correct word here. Susceptibility has to do with how susceptible an animal/person is to infection and associated disease and may be attributed to factors such as host immunologic status, host-parasite adaptation, etc. But this study does not examine the clinical aspects of infections. Consider reword to “exposure and infection”.

Line 50: What is meant by “confirm”? Is this published elsewhere?

Line 61: Toxoplasma does not cause severe health problems in most of the species it infects. Reword. Might instead write, “capable” of causing…

Line 86: Suggest alternative word to “benefit”, such as relevance or importance.

Line 144: “Toxoplasmosis” Is the disease manifested from the infection. Suggest change word to transmission.

Lines 143-147: Suggest rewording these sentences to more effectively make your point that although waterborne transmission is a significant route in terrestrial mammals, it is predicted that aquatic mammals would have a higher rate of waterborne exposure and therefore higher prevalence of infection than terrestrial mammals.

Line 148: Add “oocysts” so that sentence reads: Since T. gondii oocysts enter the aquatic system from…

Can the authors comment on the varying methods used to determine Toxoplasma infection for prevalence calculations (e.g., MAT, PCR, etc.) and how the differences in sensitivity and specificity of the different test methods used across different studies included impact the model? In particular, assay validation in wildlife species.

Figure 1: Do dark grey areas correspond to regions where no data is available? Suggest including statement in figure legend to describe the dark grey regions.

S1 and S2 Tables: Suggest including the total number of each order and taxonomic family included in the prevalence estimation.

Will the master dataset and list of publications included in the analysis be provided?

PLOS authors have the option to publish the peer review history of their article (what does this mean?). If published, this will include your full peer review and any attached files.

Reviewer #1: No

Figure Files:

Data Requirements:

Reproducibility:

References:

---

## [Editor Report · Decision Letter 1]

18 Dec 2023

Dear Dr Wilson,

We are pleased to inform you that your manuscript 'The role of species ecology in predicting Toxoplasma gondii prevalence in wild and domesticated mammals globally' has been provisionally accepted for publication in PLOS Pathogens.

Best regards,

Michael L Reese, PhD

Academic Editor

PLOS Pathogens

James Collins III

Section Editor

PLOS Pathogens

Kasturi Haldar

Editor-in-Chief

PLOS Pathogens

orcid.org/0000-0001-5065-158X

Michael Malim

Editor-in-Chief

PLOS Pathogens

orcid.org/0000-0002-7699-2064
---

## [Editor Report · Acceptance letter]

4 Jan 2024

Dear Dr Wilson,

We are delighted to inform you that your manuscript, "The role of species ecology in predicting *Toxoplasma gondii* prevalence in wild and domesticated mammals globally," has been formally accepted for publication in PLOS Pathogens.

Best regards,

Michael Malim

Editor-in-Chief

PLOS Pathogens

orcid.org/0000-0002-7699-2064